# Mobile Device-Based Train Ride Comfort Measuring System

**Yuwei Hu [1], Lanxin Xu [1], Shuangbu Wang [2], Zhen Gu [1] and Zhao Tang [1,\*]**

[1] Traction Power National Key Laboratory, Southwest Jiaotong University, Chengdu 610031, China; huyuwei2tpl@163.com (Y.H.); xinlnitia@163.com (L.X.); guzheng16399@163.com (Z.G.)

[2] Institute of Smart City and Intelligent Transportation, Southwest Jiaotong University, Chengdu 611756, China; shuangbuwang@swjtu.edu.cn

\* Correspondence: tangzhao@swjtu.edu.cn

**Abstract:** As an important train performance quality, comfort depends on vibration and noise data measured on a running train. Traditional vibration and noise measurement tools are facing challenges in terms of collecting big data, portability, and cost. With the continuous upgrade of mobile terminal hardware, the built-in sensors of mobile phones have the ability to undertake relatively complex data measurement and processing tasks. In this study, a new type of train comfort measurement system based on a mobile device is developed by using a built-in sensor to measure the vibration and noise. The functions of the developed system include the real-time display of three-way vibration acceleration, lateral and vertical Sperling indicators, sound pressure level, and train comfort-related data storage and processing. To verify the accuracy of the mobile device-based train ride comfort measuring system (DTRCMS), a comparison of test results from this system and from the traditional measuring system is conducted. The comparison results show that the DTRCMS is in good agreement with the traditional measuring system. The relative error in three-direction acceleration and Sperling values is 2~10%. The fluctuation range of the noise measured by DTRCMS is slightly lower than that of the professional noise meter, and the relative error is mainly between 1.5% and 4.5%. Overall, the study shows that using mobile devices to measure train comfort is feasible and practical and has great potential for big data-based train comfort evaluation in the future.

**Keywords:** rail vehicle; ride comfort; vibration measurement; noise measurement; Sperling index

## 1. Introduction

Train ride comfort has a great impact on the competitive advantage of rail transit and other modes of transportation. To date, evaluation of train ride comfort has mainly used the vibration and noise data of the train [1–3]. In the 1950s and 1960s, a series of comfort evaluation standards and calculation criteria were initially formed. In 1941, Sperling and Helberg of the German Railway Vehicle Research Institute gave the famous Sperling index, and this empirical formula has been widely used in many countries [4]. Chinese specifications ("Test Appraisal Method and Evaluation Standard for Dynamic Performance of Railway Locomotives" and "Code for Dynamic Strength and Dynamic Performance of High-speed Test Trains") are also mainly evaluated based on this index. The International Organization for Standardization (ISO) proposed the 1/3 octave band method and the total weighted value to calculate the comfort evaluation index [5]. The International Union of Railways (UIC) promulgated and implemented the UIC513 standard in 1994 [6]. It can be seen that there are already a few standards and methods for evaluating train comfort. However, they need to be improved in the following aspects:

1. The data used in the existing evaluation standard or methods are not enough to cover full runtime and lifecycle. This is due to the fact that traditional vibration and noise meters cannot collect such large amounts of data at any given time due to cost and portability concerns;

2. Compared with train response data, human response data are more direct in evaluating train ride comfort, but traditional vibration and noise meters, such as train sensor systems and DSP-based systems, cannot directly measure the human response data [7], and the production of the comfort index requires second processing and calculation.

Accurate results of train comfort require a large number of subjects to perform a mass of tests under the actual operating environment. Currently, without the support of low-cost and easy-to-operate measurement tools, testers often have to make a compromise to reduce the sample, which greatly decreases the accuracy of the results [8]. It can be seen that the development of portable and accurate train ride comfort measurement tools that support mass data storage plays an important role in accurately reflecting passenger comfort and train operating status and improving comfort evaluation standards.

At present, the common measurement methods include inspection vehicle monitoring, precision sensor monitoring, and on-site manual measurement [9]. These traditional measurement methods have some shortcomings, such as high maintenance cost, poor portability, and nonsupport of real-time display and data processing. In order to improve measurement efficiency, Molly et al. [10] used virtual instrument technology and designed a portable train comfort and stability testing system integrating data acquisition, processing, storage, and analysis. Using the Qt development platform, Li et al. [11] designed a set of comfort measurement systems for high-speed railway automatic driving systems which achieves a quantitative detection of the impact of the high-speed railway automatic driving curve on passenger experience. Chang et al. [12] also tried to apply a mobile phone to evaluate train vibration comfort and initially verified its feasibility and measurement accuracy. Although these system instruments can complete the measurement of noise or train vibration and the evaluation of comfort, the sensor of most designs is separated from the control display interface. It is a single-function type with low portability.

In recent years, with the continuous progress in operating systems, core processors, and sensing devices, the functions of smartphones have become increasingly powerful and have been preliminarily applied in non-destructive testing, airport noise measurement, and other fields [13–18]. Shiferaw et al. [19] used smartphone sensors to measure traffic-induced ground vibration and grasp ground health in real time. Saurabh Garg et al. [20] used smartphones to record and analyze the car noise of various high-speed railway systems and to perform data analysis on the operating noise in a train passenger compartment. Partridge et al. [21] used a sinusoidal excitation filter to calibrate the acceleration sensor of a smartphone and analyzed the flattest road in an ambulance through the massive big data collected by the mobile phone. Liu et al. [11] explained the origin of vibration and noise in railway trains in more detail and summarized the technical standards and common methods for a series of studies on noise prediction, measurement analysis, and noise reduction in railway trains. Liu [22] used mobile phone sensors to measure the vibration acceleration data of vehicles passing through bridges and realized the functions of data saving, setting sampling frequency, and real-time data display.

In this paper, a new type of train ride comfort measurement system is proposed based on a mobile terminal with built-in sensors. Not only are the functions of real-time display of three-way vibration acceleration, lateral and vertical Sperling indicators, sound pressure level, and data storage and processing realized, but measurement accuracy is also ensured. The accuracy and usability of DTRCMS are verified by test comparisons with professional measurement equipment.

## 2. Train Ride Comfort Measurement System

The system function is divided into two modules, namely, the vibration measurement module and the noise measurement module. The built-in sensor and built-in microphone of a mobile phone are used as the system input; the mobile phone interface is used as the output of the system, and data correlation processing is performed, including algorithm processing of acceleration and noise, real-time data storage processing, etc. [23]. The overall use case diagram of the system is shown in Figure 1:

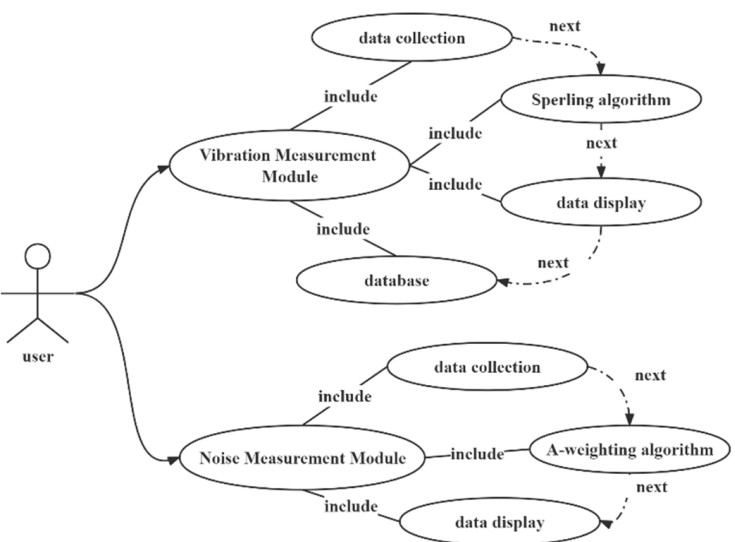

**Figure 1.** The overall use case diagram of the system.

The functions of the two modules are composed of four parts: data acquisition, data processing, data display, and data storage and evaluation. The overall implementation process and functions of the system are shown in Figure 2.

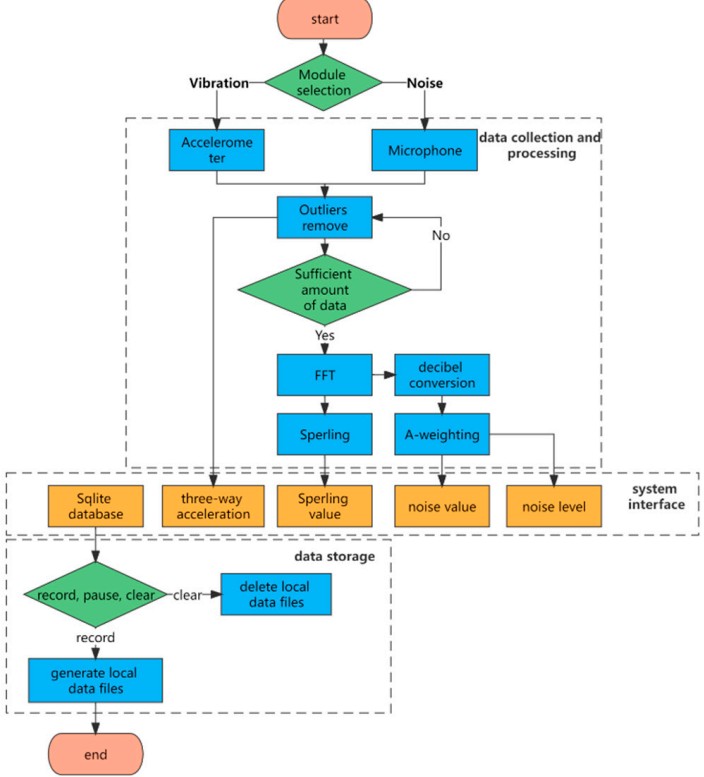

**Figure 2.** The overall flow of the system.

## 2.1. Data Acquisition

Data acquisition includes acceleration and noise. For collecting acceleration data, the built-in linear acceleration sensor was chosen to avoid the influence of gravity and to prepare for the calculation required by the Sperling index. At the same time, an appropriate sampling frequency should be adopted to prevent the frequency from being too fast to display real-time data unstably. This system selected the SENSOR_DELAY_GAME mode (50 Hz), which is one of the four sampling frequency modes of Android.

For collecting noise data, the Android system provides developers with the MediaRecorder class, which can realize an audio recording function. By creating an object of the MediaRecorder class and calling its corresponding method, the recording of audio and the acquisition of noise amplitude are realized.

### 2.2. Data Processing

Data processing includes acceleration processing and noise processing. The acceleration processing part includes the fast Fourier transform (FFT) and the Sperling algorithm. The noise processing part includes the conversion of the sound amplitude to the decibel value, the A-weighting algorithm, and the calculation of the final average decibel value of the entire measurement process.

### 2.2.1. FFT

FFT is an algorithm based on the characteristics of parity symmetry and opposite signs of virtual and real after sampling of the time-domain signal by discrete Fourier transform (DFT).

### 2.2.2. Equivalent Continuous A-Weighting

In the field of measuring noise, simply measuring decibels is no longer enough to describe the feeling that is closer to what the human ear hears. In order to reflect the high sensitivity of the human auditory system around the frequency of 3 kHz and the low sensitivity at the frequency of 60 Hz, the usual processing method is to use the A sound level weighting curve to weight the signals of different frequencies. Its formula is:

$$A(f) = 20lg\left[\frac{f_4^2 f^4}{(f^2 + f_1^2)(f^2 + f_2^2)^{\frac{1}{2}}(f^2 + f_3^2)^{\frac{1}{2}}(f^2 + f_4^2)}\right] - A_{1000}, \tag{1}$$

where $f$ is the frequency of the input signal; $A_{1000} = -2.000$ dB is a normalization constant expressed in decibels; $f_1 = 20.6$ Hz; $f_2 = 107.7$ Hz; $f_3 = 737.9$ Hz; and $f_4 = 12194.0$ Hz.

The A-weighted frequency weighting coefficient $\alpha_A(f)$ can be obtained from the following formula:

$$\alpha(f) = \frac{f_4^2 f^4}{(f^2 + f_1^2)(f^2 + f_2^2)^{\frac{1}{2}}(f^2 + f_3^2)^{\frac{1}{2}}(f^2 + f_4^2)}, \tag{2}$$

The A-weighting coefficient in the frequency domain can be obtained as:

$$X_A(k) = \alpha_A(f_k)X(k), \tag{3}$$

For discontinuous and unstable noises such as urban vehicle traffic noise and train running noise, a single A-weighted sound level cannot be accurately reflected. Therefore, this system introduces the equivalent continuous A sound level, which uses the mean of the noise energy to evaluate the impact of noise on people according to the time-average method.

According to Parseval's theorem, the total energy of the same signal in the time domain and the total energy in the frequency domain are always the same, that is:

$$\sum_{n=0}^{N-1}|x(n)|^2 = \frac{1}{N}\sum_{k=0}^{N-1}|X(k)|^2, \tag{4}$$

where $N$ is the number of sampling points; $x(n)$ is the discrete-time domain signal obtained by sampling the signal; and $X(k)$ is the Fourier transform corresponding to $x(n)$.

By using Equation (4), the average energy of the A-weighted signal can be obtained, and the corrected form is:

$$\overset{\wedge}{P} = \sqrt{\frac{1}{N}\sum_{n=0}^{N-1}|x(n)|^2} = \sqrt{\frac{1}{N^2}\sum_{k=1}^{N-1}|X_A(k)|^2},$$  (5)

Then, according to the sound level calculation formula, the A-weighted sound level can be obtained as:

$$L_A = 20\,lg\left(\frac{\overset{\wedge}{P}}{2\times 10^{-5}}\right) + 2,$$  (6)

### 2.2.3. Sperling Index Algorithm

The Sperling index is an evaluation index, summarized by Sperling et al. [24], in which vibration data are measured through a large number of experiments with human physiological sensation. It reflects the running quality of the vehicle itself and the riding comfort of passengers. Its formula is:

$$W = 0.896\,\sqrt[10]{\frac{a^3}{f}F(f)},$$  (7)

where $a$ is the measured vehicle vibration acceleration (unit: cm/s$^2$); $f$ is the frequency of the acceleration signal (unit: Hz); and $F(f)$ is the frequency weighting coefficient.

Studies have shown that the human body has very different sensitivities to vibration in different frequency bands [25]. In vertical vibration, the human body is most sensitive to vibrations at frequencies from 4 to 8 Hz; in longitudinal vibration, the human body is more sensitive to vibrations below 2 Hz. According to GB5599-2019 "Railway Vehicle Dynamic Performance Evaluation and Test Qualification Specification", the frequency weighting formula of the Sperling index is shown in Table 1:

**Table 1.** The frequency weighting formula of the Sperling index.

| Lateral Vibration/Hz | | Vertical Vibration/Hz | |
|---|---|---|---|
| 0.5~5.4 | $F(f) = 0.8f^2$ | 0.5~5.9 | $F(f) = 0.325f^2$ |
| 5.4~26 | $F(f) = 650/f^2$ | 5.9~20 | $F(f) = 400/f^2$ |
| >26 | $F(f) = 1$ | >20 | $F(f) = 1$ |

$W$, obtained from formula (7), is the value at a single frequency $f$. In practice, the vibration contains acceleration values of different frequencies. After performing the Fourier transform, it is brought into formula (8) to obtain the value of each group of frequencies. After the $W$ value, the final Sperling index can be calculated by the formula:

$$W = \sqrt[10]{W_1{}^{10} + W_2{}^{10} + W_3{}^{10} + \cdots W_n{}^{10}},$$  (8)

where $W_i$ is the Sperling index at the $i$-th frequency.

### 2.3. Data Display

Data presentation types should be as diverse as possible in order to enhance the visual performance of the user interface. The interface of the vibration measurement module mainly includes functions such as time display, curve display, real-time data display, calculation of Sperling index value, and database upload. The overall layout adopts a linear layout. The interface of the noise measurement module mainly includes a large disc display, real-time data value display, curve display, and related keys. The overall layout adopts a constraint layout, which is convenient. The user interface is shown in Figure 3.

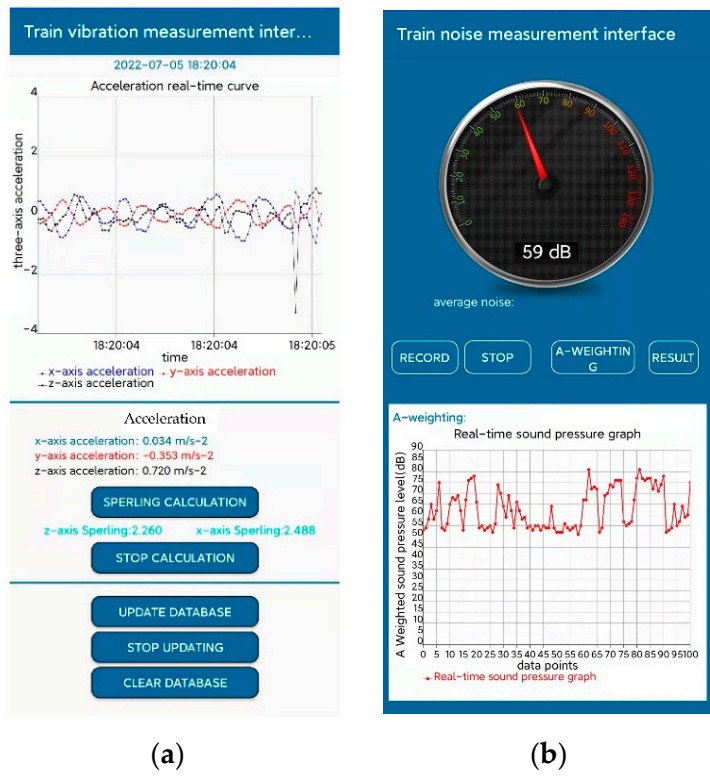

(**a**)                    (**b**)

**Figure 3.** The user interface of the DTRCMS: (**a**) the vibration measurement interface; (**b**) the noise measurement interface.

### 2.4. Data Storage and Evaluation

In order to facilitate the acceleration data and Sperling comfort index data in the vibration module for further data analysis on the computer in the later stage, the data saving function was added to the vibration module area. SQLite was selected for data storage; it is a built-in lightweight database for mobile phones, convenient for operation, and it can record various real-time data at the same time.

In the noise measurement module, the result evaluation interface was introduced. The sound pressure level obtained during the measurement period is calculated to obtain the average decibel value (the average equivalent sound pressure value) during the measurement period, and the value is generated through the result button event and passed into the resulting interface. The users can select the running line (high-speed rail or subway) to get whether the noise value during this period exceeds the noise standard limit.

## 3. System Test and Application

In order to verify the feasibility of the DTRCMS, professional noise and vibration measuring instruments were used to compare with the DTRCMS. The Sperling index calculated by the mobile phone was also compared and analyzed to verify the accuracy and reliability of the Sperling train comfort evaluation.

### 3.1. Technical Standards of Measuring Equipment

3.1.1. Professional A Sound Level Noise Meter

1.  Sampling frequency: 50 Hz, real-time measurement, fast response;
2.  A-weighted processing of ambient noise;
3.  Resolution: 0.1 dB, accuracy error: ±1.5 dB;
4.  The measurement range is 30 dB~130 dB, and the frequency response range is 31.5 Hz~8.5 kHz;
5.  The time constant is 125 ms, and the time weighting is fast.

### 3.1.2. Professional Vibration Accelerometer

6.  Integrated high-precision acceleration sensor and gyroscope;
7.  The acceleration measurement range is ±16 g, and the accuracy is 0.0001 g;
8.  The data output frequency is 100 Hz; the interface adopts serial TTL communication, and the baud rate is 115,200.

### 3.1.3. Honor 9X Mobile Phone Measuring Instrument

9.  CPU Kirin 810 GPU Turbo3.0;
10. Built-in sensors: acceleration sensor, mobile phone microphone, gravity sensor, pressure sensor, etc.

### 3.2. System Test

The noise test conducted experiments by playing an audio file in AAC format based on the ISO standard, with a sampling rate of 44.1 kHz and a bit rate of 192 Kbps. The two measuring instruments recorded two data every second, and the experiment time was 150 s. Among them, Figure 4 shows the data comparison curve diagram of the two measuring instruments in this test, and Figure 5 shows the relative error with interval distribution.

In Figure 4, the data trends of the two curves are roughly the same. The data fluctuation of the mobile phone measuring instrument is slightly lower than the standard value of the professional instrument, and the amplitude is generally slightly lower. The reasons for generating the lower fluctuation may include the following:

1.  Hardware: the limitations of the type and quality of built-in MEMS microphones in smartphones (small size, circuit board position, dynamic range and signal-to-noise ratio responsiveness, etc.);
2.  Software: the correction method of the application, the time delay of data processing, and the influence of software running in the background;
3.  Others: the influence of various obstacles (phone protection covers, microphone openings clogged by dust), different operating systems, different mobile phone brands, and environmental vibration.

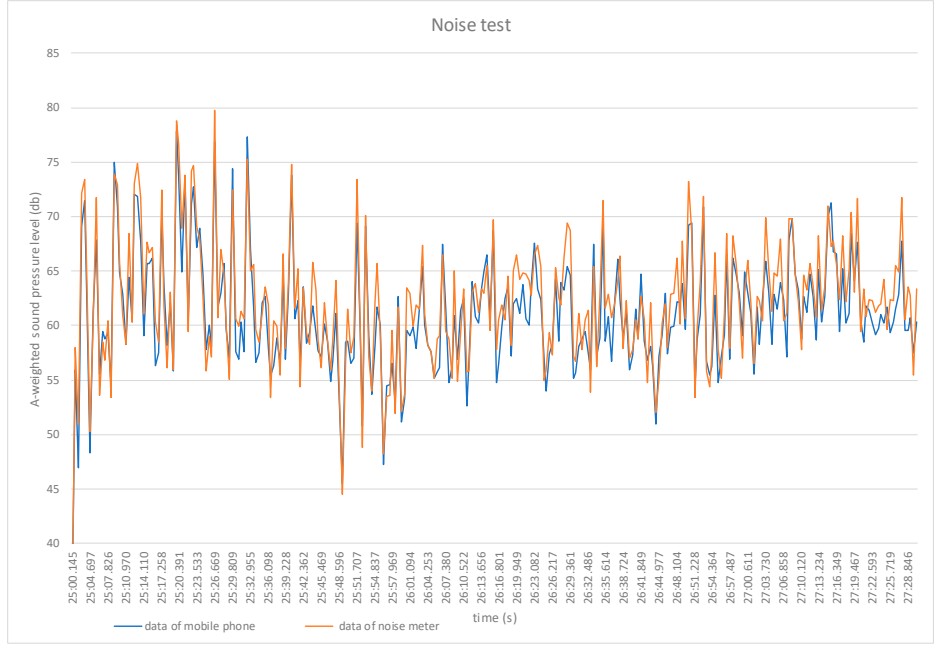

**Figure 4.** Comparison of noise data.

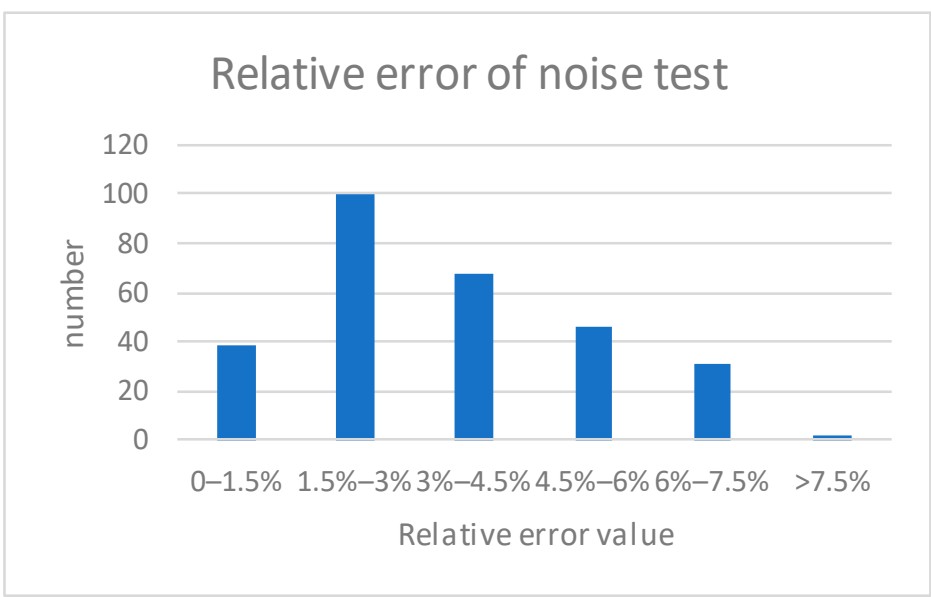

**Figure 5.** Distribution of relative error intervals.

It can be seen from Figure 5 that the relative errors of the two groups of data are mainly distributed between 1.5% and 4.5%; the maximum error is 7.83%, and the overall error is within a reasonable range.

In the acceleration test, the mobile phone measuring instrument and the professional measuring instrument were fixed on a wooden partition, and the vibration environment was created by artificial shaking. For the convenience of comparison, two data were recorded every second, and the experiment time is 120 s.

Figure 6 shows the relative error distribution interval diagram of the three-axis acceleration, and the experimental data include a total of 256 points. As shown in this figure, the relative error of the three-axis acceleration is generally distributed between 2% and 10%; the individual data error is greater than 14%, and the minimum error is less than 2%. In the three-axis data, the relative error data distributions of the y-axis and the x-axis are relatively concentrated, mainly distributed in 2~8%, and the z-axis data distribution is relatively uniform, mainly distributed in 0~10%.

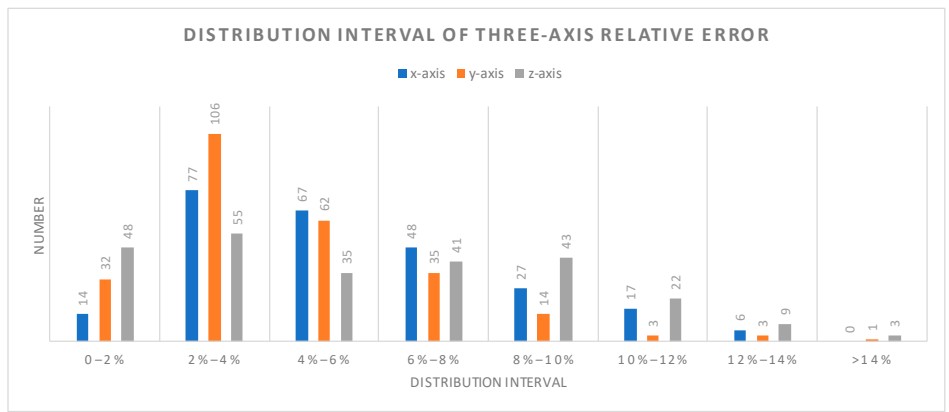

**Figure 6.** Distribution of acceleration relative error.

The Sperling index test adopted the same method as the acceleration test, and the vibration test environment was built by artificial shaking. The Sperling index of the mobile phone measuring instrument was extracted from the database, and the Sperling index of the professional acceleration instrument was calculated from its acceleration value through MATLAB. The sampling frequency of the two instruments was 50 Hz. A Sperling index

value was calculated for every 1000 acceleration data, and the experiment time was set to be 55 s. There are 10 sets of data in total, as shown in Table 2.

**Table 2.** Comparison of Sperling index values of mobile phone/professional instrument.

| No. | Horizontal Sperling | | Relative Tolerance | No. | Vertical Sperling | | Relative Tolerance |
| | Mobile Phone | Professional Equipment | | | Mobile Phone | Professional Equipment | |
|---|---|---|---|---|---|---|---|
| 1 | 2.40 | 2.37 | 1.14% | 1 | 2.56 | 2.58 | 0.66% |
| 2 | 2.41 | 2.39 | 0.96% | 2 | 2.45 | 2.49 | 1.67% |
| 3 | 3.64 | 3.49 | 4.41% | 3 | 3.28 | 3.35 | 2.12% |
| 4 | 3.49 | 3.54 | 1.24% | 4 | 3.23 | 3.23 | 0.03% |
| 5 | 3.91 | 3.79 | 3.35% | 5 | 3.26 | 3.35 | 2.66% |
| 6 | 3.44 | 3.50 | 1.66% | 6 | 2.97 | 2.93 | 1.19% |
| 7 | 3.11 | 2.97 | 4.71% | 7 | 2.94 | 2.86 | 2.62% |
| 8 | 3.06 | 2.99 | 2.14% | 8 | 3.08 | 3.10 | 0.49% |
| 9 | 3.03 | 2.96 | 2.43% | 9 | 3.08 | 3.10 | 0.87% |
| 10 | 3.11 | 3.15 | 1.07% | 10 | 3.05 | 2.99 | 1.97% |

In Table 2, there is little difference between the Sperling index value calculated by the mobile phone and the Sperling index value calculated by MATLAB; i.e., the lateral error is less than 5%, and the vertical error is less than 3%.

The data comparison shows that there is still a certain error between the mobile phone and the professional tester. The reasons for generating the above errors may include the following:

1. On the whole, due to the limitations of mobile phone cost, sensor volume, and built-in location, the measurement accuracy and sampling performance of the built-in acceleration sensor and microphone in mobile phones are not as good as professional equipment, but the gap is not obvious;

2. The computing power of the mobile phone CPU is uneven, which leads to a certain delay in the A-weighting calculation and Sperling value calculation. Subsequent cloud computing can effectively solve this problem by decomposing computing tasks into the cloud;

3. During the experiment, the positions of the two devices cannot be completely consistent, resulting in different forces, relative jitter, or non-unique variables. In the follow-up, big data processing methods can be used to filter out human factors;

4. The test time is relatively short, and the number of tests is small (all tests are short-term tests of several minutes, and the number of measurements is small; differences between different mobile phones are not considered), which leads to limited test data and causes certain statistical errors.

The overall experimental results show that the relative errors are in a normal range, and it is feasible to use the mobile phone measuring instrument as a train comfort measurement tool in order to reflect train comfort.

### 3.3. System Application

Taking Chengdu Metro Line 6 as the test train line for the DTRCMS application, the noise measurement module was applied between Wangcongci Station and Shuxin Avenue Station; the vibration of standing posture was measured between Shuxin Avenue Station and Xipu Station, and the vibration of sitting posture was measured between Tianyu Road Station and Wangcongci station. The test roadmap is shown in Figure 7.

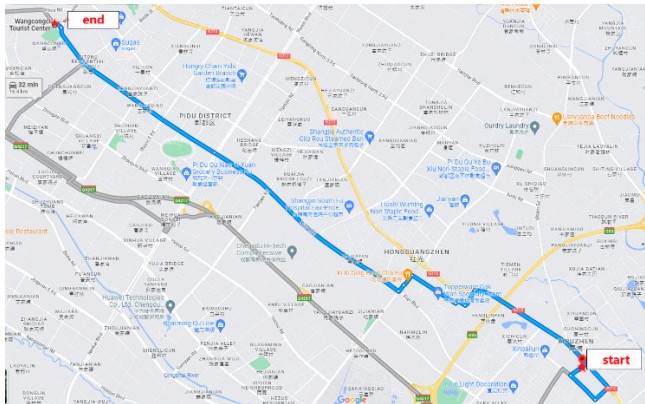

**Figure 7.** Test roadmap.

The sound pressure levels during the noise measurement process are shown in Figure 8. The data basically remained between 65 and 75 decibels. The decibel value of the running train was in the range of 70 to 75 decibels. When the train stops and starts, the noise can reach 80 decibels. According to the GB14892-2006 (Urban Rail Transit Vehicle Noise Limits and Measurement Methods) issued by China, the noise limit of subway passenger compartments is 83 decibels, and the noise measured by DTRCMS did not exceed this limit in the whole test process.

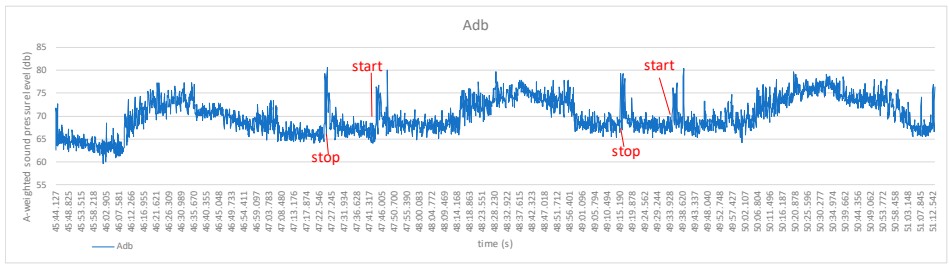

**Figure 8.** Decibels in the test process.

In the application of the vibration measurement module, the mobile phone was placed horizontally on the seat and the floor to simulate the two passenger states of sitting and standing, respectively. The x-axis direction of the mobile phone points towards the front of the train, as shown in Figure 9.

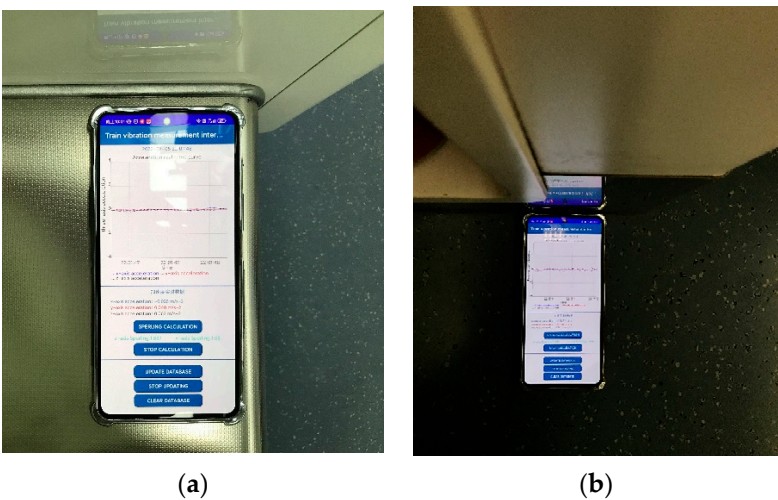

**(a)**                                                    **(b)**

**Figure 9.** Application site of vibration measurement: (**a**) the mobile phone placed on the seat; (**b**) the mobile phone placed on the floor.

The measurement time of the standing posture vibration measurement process was 12 min; the measurement time of the sitting posture vibration measurement process was 15 min. Its three-axis acceleration curve is shown in Figure 10. The acceleration fluctuations of the two states were similar. In the subway operation stage, the acceleration fluctuations of the three axes were obvious, and they were all in the range of $-0.5{\sim}0.5$ m/s$^2$. When the subway stopped and started, the x-axis acceleration increased sharply, and the maximum value reached 1.5 m/s$^2$. During the stop phase of the subway, the three-direction acceleration changed smoothly.

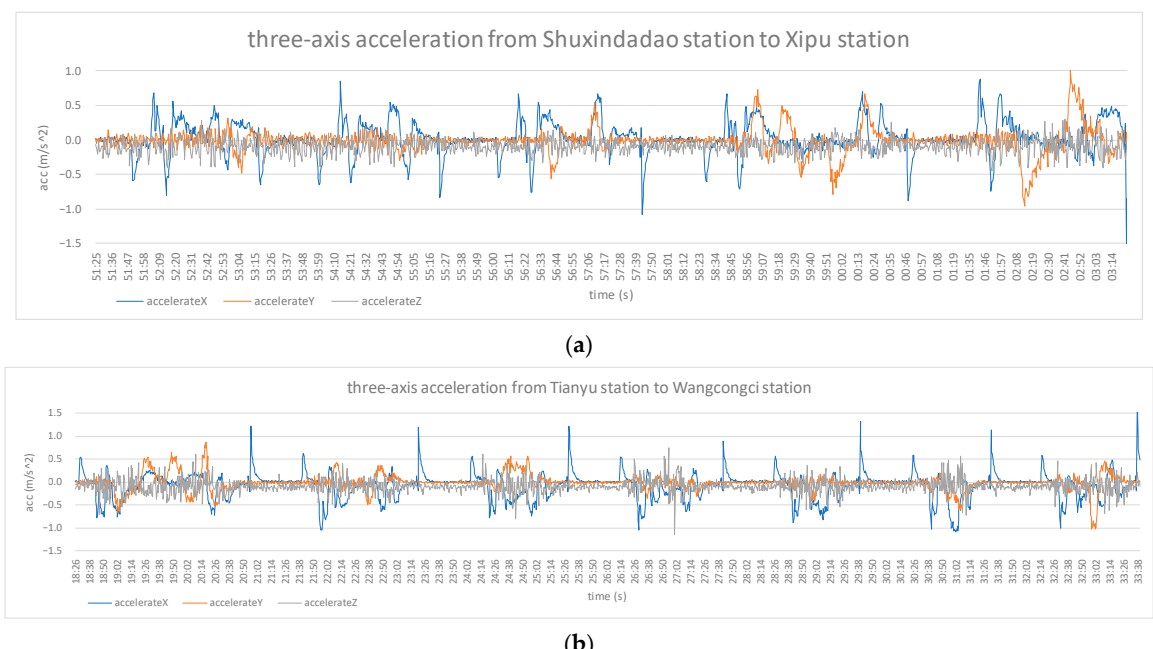

(**a**)

(**b**)

**Figure 10.** Three-way acceleration: (**a**) three-axis acceleration from Shuxin Avenue station to Xipu station; (**b**) three-axis acceleration from Tianyu station to Wangcongci station.

Horizontal and vertical Sperling indicators corresponding to the two states are shown in Figure 11, where the value of the vertical Sperling indicator fluctuated between 1.5 and 2, and the value of the horizontal Sperling index fluctuated around 3 (the value was around 3.5 during subway operation and around 1.5 during the stops). According to the Sperling rating table of locomotives and rolling stock in China, during the whole measurement process, for the horizontal Sperling index value, the subway operation stage is scored as "pass" or "good", due to the more obvious vibration in individual periods, and in the stop phase, the score is "excellent". For the vertical Sperling index value, the whole process is relatively stable, and the score is "excellent". The measured data of the two states are roughly consistent with actual human experience.

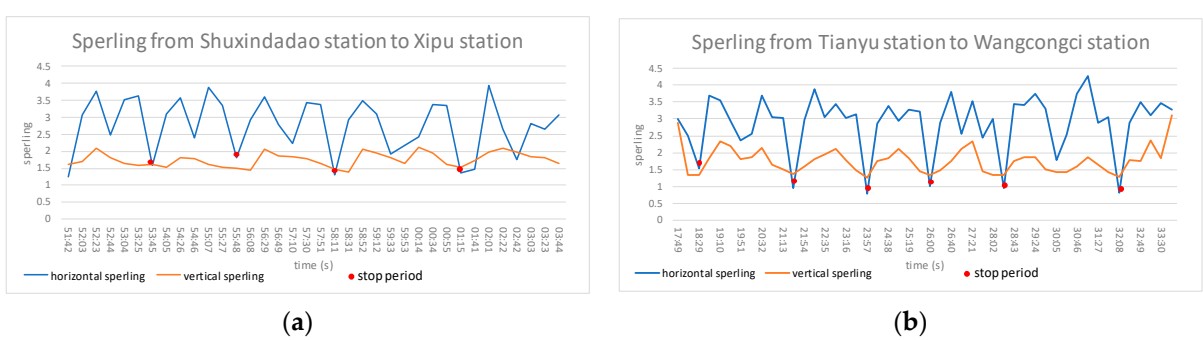

(**a**)                                      (**b**)

**Figure 11.** Sperling index value: (**a**) Sperling from Shuxin Avenue station to Xipu station; (**b**) Sperling from Tianyu station to Wangcongci station.

## 4. Conclusions

Based on smartphone mobile devices, this study develops a new portable train comfort measurement system (DTRCMS). The availability, accuracy, and reliability of the system are verified by comparison with a professional test system in an on-board test. The main conclusions are as follows:

1.  Mobile devices can realize multi-module algorithm integration to meet real-time comfort assessment and visual presentation. The FFT, Sperling algorithm, A-weighting algorithm, and other calculation processes were integrated into the system, and noise and vibration data processing could be performed in real time.
2.  The measurement accuracy of the built-in sensor in the mobile phone was verified by a comparative test. The relative error of the noise test data was mainly distributed in 1.5~5%. The overall trend of the three-way acceleration values was the same, and the relative error was 2~10%. The difference in the Sperling index was small, and the relative error was less than 5%.
3.  The availability and reliability of the DTRCMS were verified by an on-board application on Chengdu Metro Line 6. During the application process, the decibel value basically remained between 65 and 75 db, which is consistent with the actual listening experience. The Sperling value was between 2 and 3, and the measured data of the two states were roughly consistent with actual human experience.

In summary, the DTRCMS not only provided almost all train ride comfort-related functions, such as noise and vibration measuring and Sperling index calculation, but it is also more convenient and faster than a traditional test meter. In addition, the challenges in the existing riding comfort evaluation standards and methods were discussed, and an optimization direction was introduced.

## 5. Future Works

The DTRCMS provides a more convenient and faster tool for train ride comfort measurement. In the error results of the comparison test, the DTRCMS is not far from professional equipment in data measurement and calculation. In practical applications, the DTRCMS's result are similar to passengers' feelings, which shows a certain feasibility. However, due to the limitations of mobile phone hardware compared with professional meters, there is still room for optimization in terms of improving accuracy. Currently, in the era of big data, follow-up research can further upgrade and improve the DTRCMS by uploading the measurement data to a big data cloud platform and evaluating comfort through massive amounts of data collected by the DTRCMS. In this way, current train comfort assessment data can be quickly obtained in real time; train running quality can be accurately reflected; and even train running tracking status can be monitored at the same time.

**Author Contributions:** Conceptualization, Z.T. and Y.H.; methodology, Y.H.; software, Y.H. and Z.G.; validation, Y.H.; formal analysis, Y.H.; investigation, Y.H.; resources, L.X.; data curation, L.X.; writing—original draft preparation, Y.H. and L.X.; writing—review and editing, S.W. and Z.T.; visualization, Y.H.; supervision, Y.H.; project administration, Z.T.; funding acquisition, Z.T. All authors have read and agreed to the published version of the manuscript.

**Funding:** This research was funded by "Network Collaborative Manufacturing and Smart Factory" of the National Key R&D Program, grant number 2020YFB1711402.

**Institutional Review Board Statement:** Not applicable.

**Informed Consent Statement:** Not applicable.

**Data Availability Statement:** Not applicable.

**Conflicts of Interest:** The authors declare no conflict of interest.

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
