# Peer review of "Mobile Device-Based Train Ride Comfort Measuring System"

_applsci, doi:10.3390/app12146904_

Round 1

Reviewer 1 Report

This paper proposes a novel type of train ride comfort evaluation system based on a mobile terminal with built-in sensors. The paper is well written and well organized. However, some points should be taken into consideration before acceptance:

1-The authors said that “In this paper, a new type of train ride comfort measurement system is proposed 89 based on the mobile terminal with built-in sensors.”, however, they did not show any comparative study with the old types to prove and evaluate the effectiveness of their proposed system.

 2-  In line 58 to 60 the sentence “At present, the common measurement methods include inspection vehicle monitoring, precision sensor monitoring and on-site manual measurement. These traditional measurement methods have some shortcomings such as high maintenance cost, poor portability, and nonsupport of real-time display and data processing.” It needs a reference to support this claimed fact. 

3- The resolution of Fig. 1 should be improved.

4- The block just below the word "noise" in Fig. 2 "Microphone" should be in the same line.

5-Fig. 6 should be enlarged to be clear as in Fig. 5.

6-In all the figures, the unit of the axis title should be mentioned.

7-For Fig. 10, the font of the axis and the axis label is very small. I think they should not put side by side to have the ability to enlarge them.

8-The authors said that “The computing power of mobile devices can meet …. “, but, they did not mention anything related to computing power in the results. 

9- The references are up to date. However, there are no references in the year of 2022.

Author Response

Thank you very much for your valuable comments and suggestions on this paper. I have further revised the paper based on your valuable suggestions, and please find the corresponding responses below:

1-The authors said that “In this paper, a new type of train ride comfort measurement system is proposed 89 based on the mobile terminal with built-in sensors.”, however, they did not show any comparative study with the old types to prove and evaluate the effectiveness of their proposed system.

Response: In view of the time and cost of the experiment, this study only conducted a comparison experiment between the mobile based measurement system and the handheld noise meter and professional vibration sensor. More comparative studies will be conducted in our following work.

2-In line 58 to 60 the sentence “At present, the common measurement methods include inspection vehicle monitoring, precision sensor monitoring and on-site manual measurement. These traditional measurement methods have some shortcomings such as high maintenance cost, poor portability, and nonsupport of real-time display and data processing.” It needs a reference to support this claimed fact.

Response: According to your suggestion, we have added a corresponding reference [10] to support the fact of this claim.

3-The resolution of Fig. 1 should be improved.

Response: According to your suggestion, we have replaced it with a higher resolution photo.

4-The block just below the word "noise" in Fig. 2 "Microphone" should be in the same line.

Response: Thanks for your suggestion, we have revised it.

5-Fig. 6 should be enlarged to be clear as in Fig. 5.

Response: According to your suggestion, we have adjusted it.

6-In all the figures, the unit of the axis title should be mentioned.

Response: Thanks for your suggestion, we added the unit to the axis of all figures in the paper..

7-For Fig. 10, the font of the axis and the axis label is very small. I think they should not put side by side to have the ability to enlarge them.

Response: According to your suggestion, we have adjusted it.

8-The authors said that “The computing power of mobile devices can meet …. “, but, they did not mention anything related to computing power in the results.

Response: According to your suggestion, we have revised Conclusion 1, from the perspective of multi-module algorithm integration.

9-The references are up to date. However, there are no references in the year of 2022.

Response: We have added some references in the year of 2022.

Reviewer 2 Report

The article presents an application of mobile devices to train passenger comfort, which interests the railway technical and scientific community. 

Here are some specific comments:

In general, avoid using terms such as "noise decibel", which is not precise. I suggest using "sound pressure level" or "A-weighted sound pressure level" instead.

Page 2, lines 69-72. Please check this phrase, which is not clear.

Page 6, lines 192-193. Does "average level" indicate the equivalent sound pressure level?

Page 6, lines 198-199. You refer to "professional noise and vibration measuring instruments". It would be helpful to indicate the technical standards to which the instruments must comply.

Page 7, lines 219-221. Please specify the characteristics of the audio signal used for the acoustic system test. Also, please specify the criteria for selecting that specific audio signal.

Page 7, Figure 4. Please provide a more precise definition of the displayed data in dB: are they A-weighted sound pressure level values? Also, please specify the time constant used by the professional SPL meter (e.g. fast?).

Page 7, lines 228-230. can you provide a tentative explanation for the lower fluctuation of the mobile device noise data? Could this be related to the time response, or to the dynamic range of the built-in microphone?

Page 8, lines 234-236. Please provide a more detailed description of the vibration test. In particular, the features of the wooden partition and of the shaker used in the test should be specified, as well as the characteristics of the shaker input signal (acceleration amplitude, frequency range, etc.).

Page 9. Lines 261-263. Please expand item 1. of the list (accuracy and performance of the mobile phone sensors). Also, please expand item 4. (duration of tests).

Page 9, figure 8. Please better define the acoustic descriptor displayed in the graph (e.g. A-weighted SPL, or short-Leq, etc.).

Page 10, Figure 9. The two images are not very clear: can you provide better pictures?

Page 10, Figure 10. The two graphs are difficult to read. I suggest paging the graphs one on top of the other, rather than side-by-side.

Page 11, lines 350-351. Please better specify the meaning of the statement "the DTRCMS has certain accuracy and feasibility in data measurement and comfort evaluation".

Author Response

Thank you very much for your valuable comments and suggestions on this paper. We have further revised the paper based on your valuable suggestions, please find the corresponding response below:

In general, avoid using terms such as "noise decibel", which is not precise. I suggest using "sound pressure level" or "A-weighted sound pressure level" instead.

Response: We improved the expression you mentioned above by changing "noise decibel" to "sound pressure level", as well as others.

Page 2, lines 69-72. Please check this phrase, which is not clear.

Response: According to your suggestion, we have re-edited this phrase.

Page 6, lines 192-193. Does "average level" indicate the equivalent sound pressure level?

Response: Yes, it is. “average level” indicates the average equivalent sound pressure value over the entire measurement time. We have added a sentence to explain it in section 2.4.

Page 6, lines 198-199. You refer to "professional noise and vibration measuring instruments". It would be helpful to indicate the technical standards to which the instruments must comply.

Response: Many thanks for your suggestion. The technical standards are described in section 3.1.

Page 7, lines 219-221. Please specify the characteristics of the audio signal used for the acoustic system test. Also, please specify the criteria for selecting that specific audio signal.

Response: The audio file used in the noise test is a music file in AAC format based on the ISO standard, with a sampling rate of 44.1kHz and a bit rate of 192Kbps. It is only used for comparison between mobile phones and professional equipment. You can find it in section 3.1.1 and section 3.2.

Page 7, Figure 4. Please provide a more precise definition of the displayed data in dB: are they A-weighted sound pressure level values? Also, please specify the time constant used by the professional SPL meter (e.g. fast?).

Response: According to your review, we have provided a more precise definition (A-weighted sound pressure level values(db)) of the displayed data in dB. We have added the time constant information (fast, 125ms) used by professional SPL meters in section 3.1.1

Page 7, lines 228-230. can you provide a tentative explanation for the lower fluctuation of the mobile device noise data? Could this be related to the time response, or to the dynamic range of the built-in microphone?

Response: There are several reasons for the lower fluctuation of mobile device noise data, which we divide into three aspects and briefly describe them in the paper. It can be found in section 3.2.

  1. Hardware: the limitations of the type and quality of built-in MEMS microphones in smartphones (small size, circuit board position, dynamic range and signal-to-noise ratio responsiveness, etc.)
  2. Software: the correction method of the application (the application needs to be calibrated to read accurate data), the time delay of data processing, and the influence of software running in the background.
  3. Others: the influence of various obstacles (phone protection covers, microphone openings clogged by dust), different operating systems, different mobile phone brands, and environmental vibration.

Page 8, lines 234-236. Please provide a more detailed description of the vibration test. In particular, the features of the wooden partition and of the shaker used in the test should be specified, as well as the characteristics of the shaker input signal (acceleration amplitude, frequency range, etc.).

Response: Due to the lack of professional shaker, the method used in this vibration test is to fix professional equipment and mobile phones on a wooden board and shake the wooden board artificially to provide a vibration environment. Therefore, the characteristics of the shaker input signal (acceleration amplitude, frequency range, etc.) are difficult to confirm.

Page 9. Lines 261-263. Please expand item 1. of the list (accuracy and performance of the mobile phone sensors). Also, please expand item 4. (duration of tests).

Response: According to your suggestion, ’we have expanded item 1 and item 4.

Page 9, figure 8. Please better define the acoustic descriptor displayed in the graph (e.g. A-weighted SPL, or short-Leq, etc.).

Response: According to your suggestion, we modified “db” to be “A-weighted sound pressure level”.

Page 10, Figure 9. The two images are not very clear: can you provide better pictures?

Response: We have replaced the two photos of Figure 9.

Page 10, Figure 10. The two graphs are difficult to read. I suggest paging the graphs one on top of the other, rather than side-by-side.

Response: Thanks for your suggestion, we have adjusted it.

Page 11, lines 350-351. Please better specify the meaning of the statement "the DTRCMS has certain accuracy and feasibility in data measurement and comfort evaluation".

Response: Based on your suggestion, we have specified the meaning of the statement. In the error results of the comparison test, the performance of DTRCMS is close to professional equipment in data measurement and calculation; In practical applications, results of DTRCMS are similar to passengers’ feelings, which has a certain feasibility. It can be found in section 5.

Round 2

Reviewer 1 Report

Dear authors

Thank you for addressing all the comments. However, all the references should be mentioned in the text of the paper. For example, reference 2 was not mentioned in the text.

Best regards

Author Response

Thank you very much for your suggestions and comments. According to your suggestions, We checked and reordered references in the paper.

Reviewer 2 Report

The authors have adequately replied to the comments in the revised version of the article.

Author Response

We greatly appreciate your comments and suggestions on this paper. Thank you very much.